# Optimization of Actuation Load and Shape Recovery Speed of Polyester-Based/Fe_3_O_4_ Composite Foams

**DOI:** 10.3390/ma14051264

**Published:** 2021-03-07

**Authors:** Tamem Salah, Aiman Ziout

**Affiliations:** Department of Mechanical Engineering, UAE University, AlAin 15258, United Arab Emirates; 201250499@uaeu.ac.ae

**Keywords:** actuation load, shape memory effect, shape memory polymer, glass transition temperature, discontinuity phase

## Abstract

In this research, polyester-based polymers/Fe_3_O_4_ nanocomposite foams were prepared in order to study their performance; namely shape recovery speed and actuation load. A foamed structure was obtained through a solid-state foaming process, which was studied and optimized in previous research. The optimum foaming parameters were applied in an attempt to achieve the highest foaming ratio possible. A Taguchi Map was then designed to determine the number of experiments to be conducted. The experimental results showed that the maximum actuation load obtained was 3.35 N, while optimal (fastest) recovery speed was 6.36 mm/min. Furthermore, temperature had no impact on the actuation load as long as a temperature above the T_g_ was applied. Moreover, the addition of nanoparticles reduced shape recovery speed due to discontinuity within the polymer matrix.

## 1. Introduction

Materials that exhibit shape memory effect are getting more attention, that is due to their suitability for various applications and industries. This paper focuses on the development of shape memory foam based on polyester polymer composite.

Shape memory effect (SME) is the phenomenon whereby the original shape of a material is recovered in the presence of a suitable stimulus such as heat, light, electric, or magnetic fields [1]. This has been observed in many different materials, including metallic alloys and polymers [2]. SME consists of two different processes, as shown in Figure 1: Programming process in which the material is deformed into a temporary shape, and the recovery process during which the material recovers its original shape.

SMPs and their foams are distinguished by their unique properties such as light weight, high elastic deformation [3], ease of processing, low cost, and excellent shape recovery properties. Shape memory polymers are widely used for different applications, being employed as light actuators, structural parts with a reduced size during transport, and as expandable/deployable structures. Other potential applications are in the biomedical field, and these include drug delivery, biosensors, and biomedical devices. Moreover, since polymers can become biodegradable, they can be used as short term implants when removal by surgery is hazardous [4].

Foams can be classified based on their cellular structures to open-cell foams like carbon, metallic and ceramic foams, and closed-cell foams like polyurethane and polystyrene foams. Common methods for producing foams are chemical [5] or physical [6,7,8] processes. These are complex methods, as they require the insertion of blowing agents. An emerging foaming process that does not require a chemical reaction or a physical foaming agent is known as solid-state foaming. This process was proposed and tested on epoxy polymers by Quadrini and his team [9]. The main purpose of this foaming process is to simplify the process and reduce costs.

This research is motivated by the advantage of foam compared to the solid form of the polymer. Shape memory polymer foams have advantages over normal shape memory polymers due to their low density and their ability to be compressed. Yet, a major disadvantage of the foam is its low recovery force with a reduced stiffness and mechanical strength. This research introduces the use of composites to overcome this disadvantage. Shape memory composites have greater strength and stiffness [10,11,12], as well as other special properties [13] determined by the types of filler that are added. Lisuzzo et al. [14] found that adding 10 wt% halloysite nanotubes to Mater-Bio plastic improved its elongation of the composite by 100%. Linul [15] and his team found that Aluminum microfiber improved mechanical properties of polyurethane flexible foams. Another study by Meesorn [16] proved that improved dispersion of cellulose nanocrystals (CNC) enhanced the mechanical properties of EO-EPI/CNC nanocomposites.

Previous work showed that The addition of Fe_3_O_4_ nanoparticles (NPs) caused an increase in the tensile strength of the poly(d,l-lactide) polymer [11]. Similarly, another study showed improvement in the resultant compressive strength, when Fe_3_O_4_ NPs were added to polyimide [17]. The results of previous studies motivate this research to study the effect of adding NPs to polyester-based foam. This research aims to find out the effect of adding Fe_3_O_4_ NPs to polyester-based polymer foam; specifically, the impact on the actuation load, and recovery speed of the composite foam. Polyester-based foam was introduced for the first time by Quadrini and Sque [9], and its performance under microgravity was studied by Santo and Quadrini [18]. Santo [19] conducted further development to improve the foam performance. Santo and Tedde [20] explored its application as an actuator.

The light weight and compressibility of polyester-based foam makes it suitable for space applications; this was investigated by Santo [21]. The developments and potential of this shape memory foam also motivate this research to explore further improvements.

The next section, experimental method, demonstrates the use of a Taguchi Map as a method to guide the study to find the best parameter levels as well as the best combination of process parameters that lead to the optimal objectives. Results are shown in Section 3 and discussed in Section 4. It is clear that these NPs impact the mechanical properties of their host matrix. The paper concludes by Section 5, where major findings are presented.

## 2. Experimental Method

### 2.1. Design of the Experiment

The Taguchi Map was designed using Minitab software (Minitab.v17 Minitab, LLC, State College, PA, USA) and was used to determine the effect of different parameters on the shape memory foam performance. The minimum number of experiments was ascertained via the formula below:(1)N_Taguchi  = 1 + N_v  (L−1)
where $N_{Taguchi}$ is the number of experiments conducted, $N_{v}$ is the number of parameters, and L is the number of different levels in each respective parameter. In this research, six different factors were studied in terms of their effect on the foamed sample properties. These factors were: NPs percentage, polymer type, packing pressure, holding time, foaming temperature, and foaming time. The selected factors and their levels were determined in a previous research by the authors. The orthogonal array design in this research needs a total of 7 experiments, the nearest Taguchi array design is with 8 experiments, and Table 1 below demonstrates the full map of the conducted experiments. Three replicas were created from each set of experiments.

### 2.2. Sample Preparation

The first step was preparing the Fe_3_O_4_ nanoparticles using co-precipitation method. Microscopic investigation of the prepared NPs was carried out using Scanning Electron Microscope technique (JEOL Ltd., Tokyo, Japan) as shown in Figure 2. The prepared NPs had an average diameter of 3 µm, similar average sizes were obtained by [21]. Following, the NPs were mixed with the polymer powder. The total weight of the mixture was 5 g, with and without NPs. X-ray powder diffraction XRD graphs (SHIMADZU CORPORATION, Kyoto, Japan) of both CC and Jotun Super Durable 2903^®^ (JSD) composites are preset in Figure 3 and Figure 4, respectively, and the graphs indicate the similarity between two polymers as both of them contain same percentage of NPs and are both polyester based polymers. Tablets were then prepared from the mixed powder by applying a certain packing pressure and holding time based on the Taguchi design. The mixture was filled into a stainless-steel mold and placed on a stainless-steel base. The powder was then packed gently into the mold using a stainless-steel plunger. The Carver Hydraulic Press Test System (CARVER, Inc., Wabash, IN, USA) provided the required packing pressure at a pumping speed of 1.56 mm/ s and the system can apply a maximum compression force of 30 klbs. Application of high pressure caused the powder to stick together and form a tablet with a height of 9 mm and a 20 mm diameter. Finally, the tablet was extracted from the mold by gentle hammering in order to avoid any cracking or breakage.

### 2.3. Tablet Foaming

The prepared tablets were then moved to the second processing stage; namely, the foaming stage. Tablets were placed into a thin aluminum sheet and then slowly inserted into another stainless-steel mold with an inner diameter of 20.7 mm. The mold was then inserted in an oven previously set to a predetermined temperature to assist with the foaming process. The Taguchi Map was used to determine the foaming temperature and foaming time to be applied to the tablets inside the oven. After the foaming time was over, tablets were carefully removed from the oven and cooled down for 15 min in an air environment. This foaming technique is called solid-state foaming and has only recently been proposed and tested on 3M epoxy resin by Quadrini and his team. After the foaming stage, samples had a cylindrical shape, and were extracted from the mold with via gentle hammering action to avoid cracking or breaking them, and then machined to produce a uniform shape and dimensions.

### 2.4. Shape Recovery Speed Measurements

Three replicas were prepared and tested to study the shape memory effect. The same technique as above was followed in order to obtain the foamed samples. All the foamed samples prepared, from each set of experiments, were placed in an oven previously heated to 120 °C. They were kept at that temperature for two minutes to soften their structure and make it easier to compress them. Subsequently, these samples were compressed using a scaled plunger for one minute at room temperature in an air environment to reach 50% of their original height. Samples were placed in a cylindrical mold in order to direct the motion downwards during compression. Furthermore, they were allowed to cool down to receive their temporary shape. After that, the compressed samples were placed in the oven once again to allow the shape recovery process to take place. The samples were removed once they had returned to their original height, with a high recovery ratio of 100%. See Figure 5 below.

The recovery time was recorded the moment samples started the recovery process until they reached their original height. It was noticed that the shape recovery process took place once the temperature inside the oven reached 120 °C. The recovery speed was calculated by dividing the height recovered by the recorded recovery time. The original height of the sample was 42 mm, compressed to 21 mm, and was 42 mm after recovering its shape.

### 2.5. Actuation Load Measurements

A new set of three samples for each experiment was prepared. These samples were placed in an oven for two minutes at 120 °C to soften their structure. Then, the samples were compressed to 50% of their original height using a scaled plunger and placed inside a cylindrical mold. The samples were then allowed to cool in an air environment for five minutes in order to achieve their temporary shape. After that, the samples were placed in an oven and in slight contact with a digital force gauge (DFG35 Digital Force Gauge, Omega Engineering, Inc., Norwalk, CT, USA)^®^ previously inserted inside the oven. Subsequently, the gauge was moved upwards to make a 0.0 N contact force with the compressed sample. The temperature inside the oven was then increased to 120 °C, and the samples were exposed to that temperature for one minute to allow shape recovery. Once the SME was activated, the sample started to exert a pushing force on the force gauge. Next, the temperature was gradually decreased to room temperature. The force gauge readings were obtained using MESUR Lite software in order to determine the actuation load values (refer to Figure 6 for a sample graph obtained using the force gauge data).

## 3. Results

### 3.1. Optimizing Shape Memory Effect (Recovery Speed)

Samples for this test were allowed to fully recover their original height, the recovery time and speed were recorded and calculated, in order to study the effect of different process parameters on the shape recovery properties of both the Corro-Coat PE Series 7^®^ (CC) and Jotun Super Durable 2903^®^ (JSD) materials. The results of the calculated recovery speed in mm/min, and the recovery time in seconds, alongside the standard deviations, are shown in Table 2 below.

An analysis of the Taguchi Map showed that lower NPs% resulted in faster recovery speeds. On the other hand, higher foaming temperature and a longer foaming time, recorded higher shape recovery speeds and lower recovery times. The CC foamed tablets had a tendency to recover faster than the JSD tablets. A summary of the analysis can be found in Figure 7 below.

### 3.2. Optimizing Actuation Load

The actuation load is defined as the force exerted by the sample on an object. Prepared samples were individually placed almost one millimeter away from the force gauge to make a 0.0 N contact force with the force gauge arm. Then, the temperature was increased to 120 °C, once the sample was actuated, the gauge recorded the actuation load applied over time as shown in Figure 3. When maximum load was achieved, the actuation load showed a decrement behavior, as all energy stored within the polymer chain was released. After that, the temperature inside the oven was returned back to room temperature. The results for the calculated mean value of the maximum actuation load applied by samples in each experiment, alongside the standard deviation values, can be found in Table 3 below.

The results showed that as NP%, foaming temperature, and foaming time increased, higher actuation loads were obtained. Additionally, lower packing pressure increased the actuation load. In this test, the JSD polymer was superior in performance to the CC, due to its more rigid structure. A summary of the analysis can be seen in Figure 8 below.

Figure 9 below shows the actuation load curves for every experiment. Each curve represents the average value for each experiment. It can be seen that run number eight (8) had the highest actuation load of approximately 3.35 N, while run number seven (7) was second with a maximum actuation load of 3.016 N. Runs four (4), one (1), two (2), five (5), and three (3) came third, fourth, fifth, sixth, and seventh, respectively, with average actuation loads of 2.35, 2.36, 1.66, 1.46, and 0.9 N. Run number six (6), however, recorded the lowest average actuation load of only 0.45 N. It was observed that all curves had very similar increment rates; it took every sample approximately two minutes (120 s) to reach their maximum actuation load. All samples showed an indistinguishable behavior; after they achieved their maximum actuation load, it started to decrease again, but none reached to 0 N by the end of the experiment period. The lowest actuation loads recorded were 0.25 and 0.35 N for runs three (3) and six (6), compared to maximums of 2.68 N for run number eight (8), and 2.3 N for run number seven (7).

It is also worth mentioning that the samples started to apply actuation loads when the oven temperature was between 80 °C and 90 °C, temperature had no further impact on the actuation loads of any of the samples.

#### Relationship between Actuation Load and Sample Length

A separate test was conducted on six different samples to verify if actuation load was dependent on sample length. A sample of Pure PE was prepared, and it was compressed under a pressure of 7500 lbs. with one minute holding time at a temperature of 260 °C, and was kept inside the oven for fifteen minutes. Three of the six prepared samples were machined to a height of 40 mm, while the other three were machined to 20 mm. All the samples were then compressed to 50% of their original height to be ready for the test. A t-test for equal means was conducted in order to compare the mean actuation loads of the two sets of samples. T-test, using Minitab, was used to determine if the two means were equal. The null hypothesis was that there is no difference between the two means for actuation loads with a confidence level of 90%. The alternative hypothesis suggested that both means were not equal. The formula used to arrive at the *t*-value is shown below.
(2)$$t=\frac{\bar{Y_{1}}-\bar{Y_{2}}}{\sqrt{\frac{s_{1}^{2}}{N_{1}}+\frac{s_{2}^{2}}{N_{2}}}}$$
where $N_{1}$ and $N_{2}$ are the sample sizes, $\bar{Y_{1}}$ and $\bar{Y_{2}}$ are the sample means, and $s_{1}^{2}$ and $s_{2}^{2}$ are the sample variances. The t-test results showed that there was no effect on the actuation load. A t-value of 2.62 was obtained using the formula above. The *p*-value was 0.142, as shown in Table 4 below. This is more than the acceptable significance level of α = 0.1, so the null hypothesis is accepted and the mean values for sample heights of 40 mm and 20mm were considered as equal.

## 4. Discussion

### 4.1. Optimizing Shape Memory Effect (Recovery Speed)

Shape memory effect is defined as the ability of a sample to return to its original shape after being deformed. This test was carried out on each sample to measure its shape recovery speed. The recovery speed is obtained by dividing the recovered height by recovery time. The fastest recovery speed was achieved when no NPs were added to the polymer matrix. See Table 5 below for results.

This can be attributed to the fact that the addition of NPs created a discontinuity in the polymer matrix, which caused the sample to take a longer time to recover its original height, hence the slower recovery speed. This observation is in line with observations by Genus and his team [22], who observed that silicon carbide (SiC) nanoparticles damaged the shape recovery feature of shape memory epoxy and shape memory polyurethane (SMPU). This negative impact was ascribed to the dramatic decrease of soft segment crystallinity in the SMPU. It was also noted that holding time under packing pressure had a negligible effect on recovery speed. See Table 6 below.

### 4.2. Optimizing Actuation Load

JSD polymer had a higher actuation load than CC. This can be attributed to the fact that JSD has a higher yield strength compared to CC (see Figure 10). Thus, it was able to store more energy. Moreover, foamed JSD is denser than CC (0.29 g/cm^3^ compared to 0.25 g/cm^3^), and Figure 11 demonstrates the fact that the volume of 5 g of foamed CC is larger than the volume of 5 g of foamed JSD, and hence was capable of exerting a greater actuation load upon shape recovery process. Both samples showed similar behavior to other foams produced by [15,23]. The graph could be characterized into 3 main regions: A linear elastic region at the beginning, a plateau region in the middle, and a densification area at the end. It is obvious that both polymers featured brittle matrices by the presence of a jagged line in the middle region.

The Addition of NPs had a small impact on improving the actuation load, the average actuation load for pure samples is 1.82 N, while the average for samples after adding 2% Fe_3_O_4_ is 2.06 N, this contributes to a 11% increment. It was beyond the initial expectations of this research. This result is in line with conclusions made by previous work, in which the addition of nanoparticles did not significantly affect the mechanical properties of the polymer composite [24]. Meanwhile, other studies reported better improvement [25,26]. This indicates an inconsistency which is attributed to the difference in polymers types, additives, and the foaming process.

The samples started exerting an actuation load on the force gauge once they reached a temperature between 80 °C and 90 °C. This temperature is just above the T_g_ of both polymers. The results indicated that after the T_g_ temperature has been exceeded, no effect of temperature on the actuation load can be found. Moreover, a further increase in temperature did not result in an increase in the actuation load. This is because all energy stored within the polymer chains had been released, and no more chain sliding or shape recovery occurred.

Table 7 below represents a comparison between mechanical tests operated on different polymeric materials. It is clear that the effect of NPs depends on various parameters [27], such as the properties of the host material. Our objective was to produce a shape memory polymer composite foam with high ratio, low density, fast shape recovery speed, and a high actuation load, while using the optimum process parameters using a Taguchi design method. So, our object was to improve memory effect, not the yield strength of the polymer. Our results are consistent with studies conducted on similar shape memory polymers, such as studies conducted by Yu [28] and Genus and his team [22].

## 5. Conclusions

A solid-state foaming process, with no foaming agent, was tested on two different polymers: Namely, Corro-Coat PE Series 7^®^ (CC) and Jotun Super Durable 2903^®^ (JSD) and their composites with the inclusion of Fe_3_O_4_ NPs. Tablets were prepared at different levels as indicated by the Taguchi Map designed for this research. These tablets were foamed at different levels of foaming temperature and varied foaming times. Moreover, different tests were conducted on these foamed samples to measure their shape recovery speed (mm/min) and actuation loads (N).

The results showed that the insertion of NPs into the polymer matrix did not increase the shape recovery speed, and in fact, this caused a reduction in speed. This can be attributed to the fact that NPs do not possess shape memory behaviors as part of their nature, and they caused discontinuity within the polymer matrix.

Generally, the JSD polymer matrix showed higher actuation load values compared to CC, due to its higher yield strength and density. NPs insertion increased actuation load of the JSD composite, and reduced actuation load of the CC composite, this is because density increased for JSD samples, and decreased for CC. Increment in density means that the energy received upon compression was stored in smaller volume, and thus able to release more force when it recovered its shape.

## 6. Future Work

Further pores expansion techniques can be tested for bigger pore size and better uniform size distribution. These techniques could be rotational or ultrasonic vibration.

## Figures and Tables

**Figure 1 materials-14-01264-f001:**
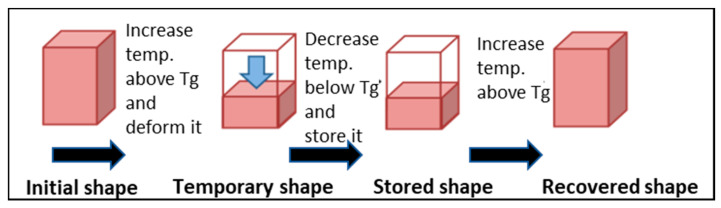
Shape memory effect steps in polymers.

**Figure 2 materials-14-01264-f002:**
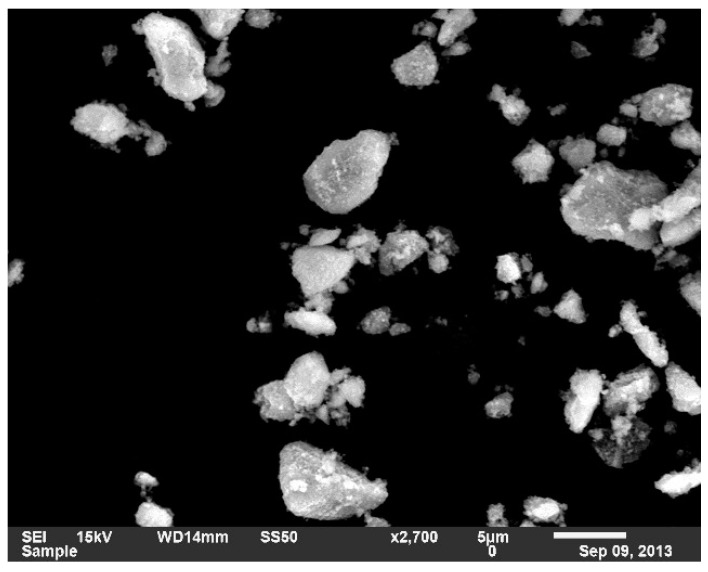
Scanning electron microscope (SEM) image of the prepared NPs.

**Figure 3 materials-14-01264-f003:**
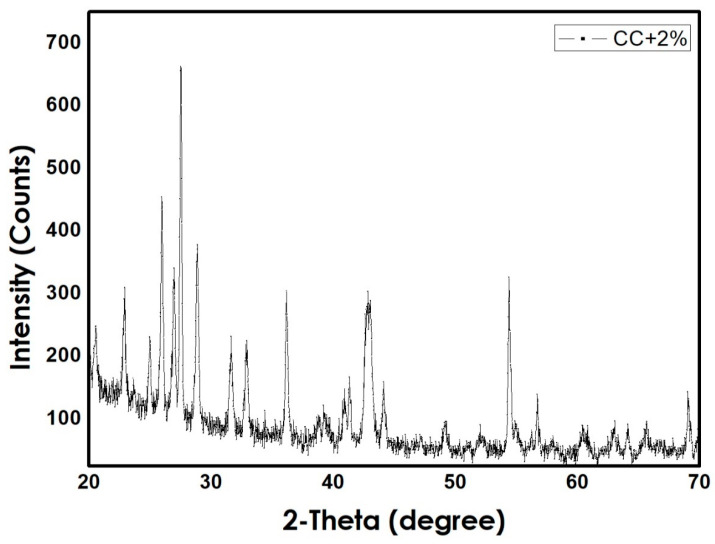
XRD pattern for CC + 2% Fe_3_O_4_ composite.

**Figure 4 materials-14-01264-f004:**
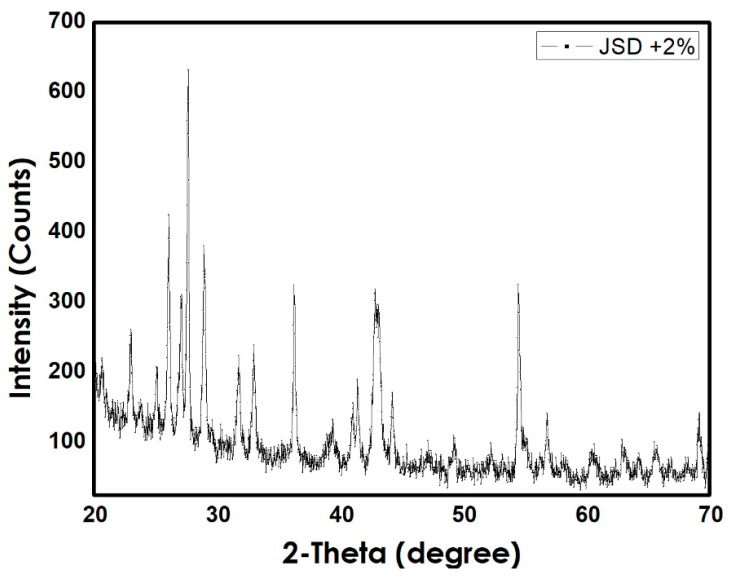
XRD pattern for JSD + 2% Fe_3_O_4_ composite.

**Figure 5 materials-14-01264-f005:**
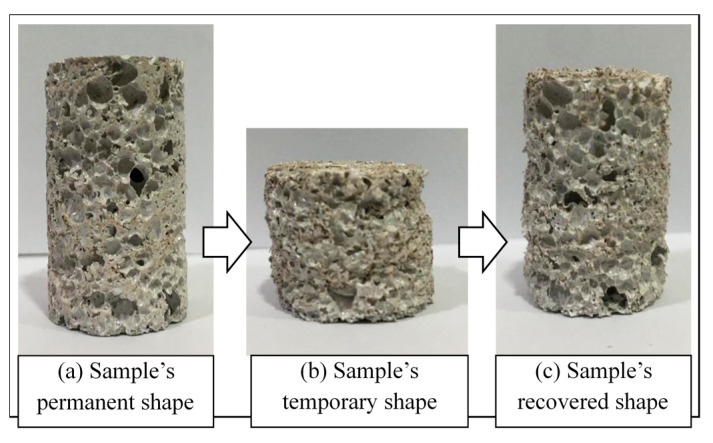
A sample just heated: Compressed to 50%: Regained original height.

**Figure 6 materials-14-01264-f006:**
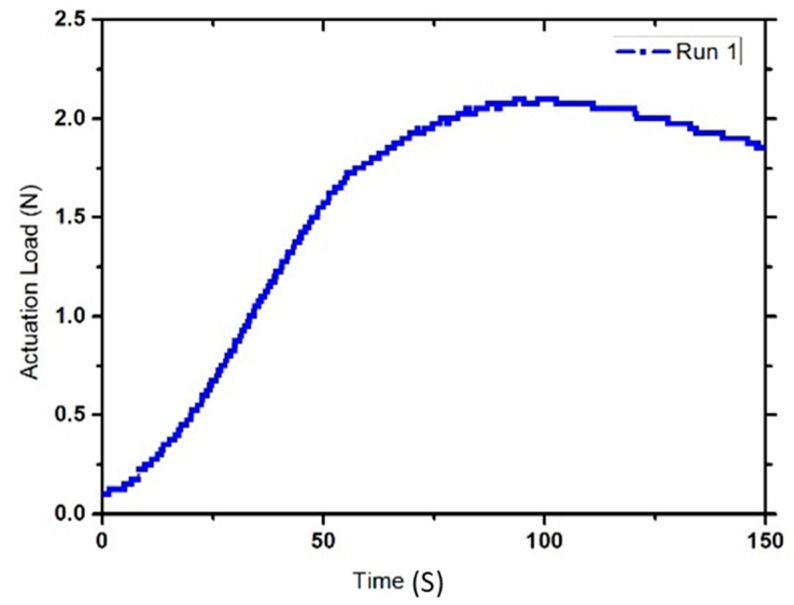
A graph generated using data extracted by MESUR Life software^®^ and the force gauge.

**Figure 7 materials-14-01264-f007:**
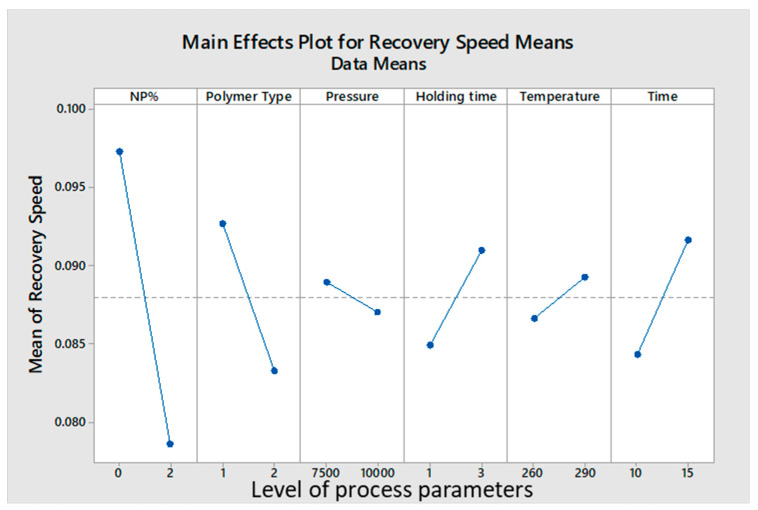
Effect of process parameters on the recovery speed of the samples.

**Figure 8 materials-14-01264-f008:**
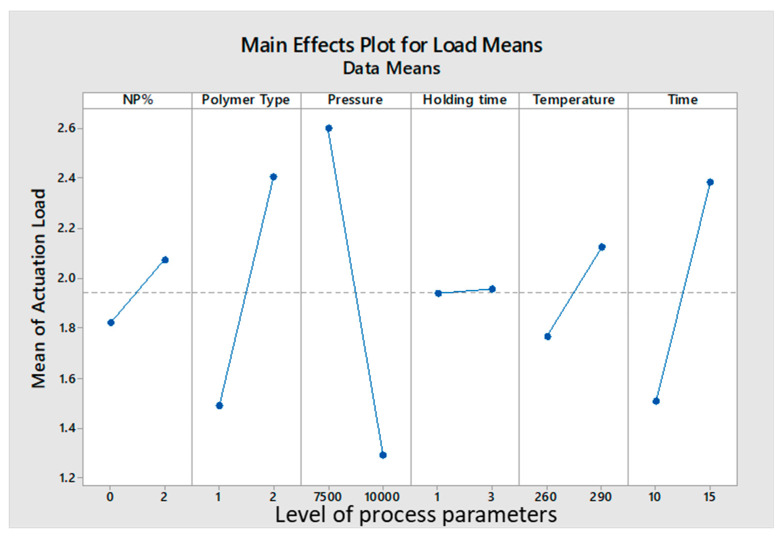
Effect of parameters on the actuation load.

**Figure 9 materials-14-01264-f009:**
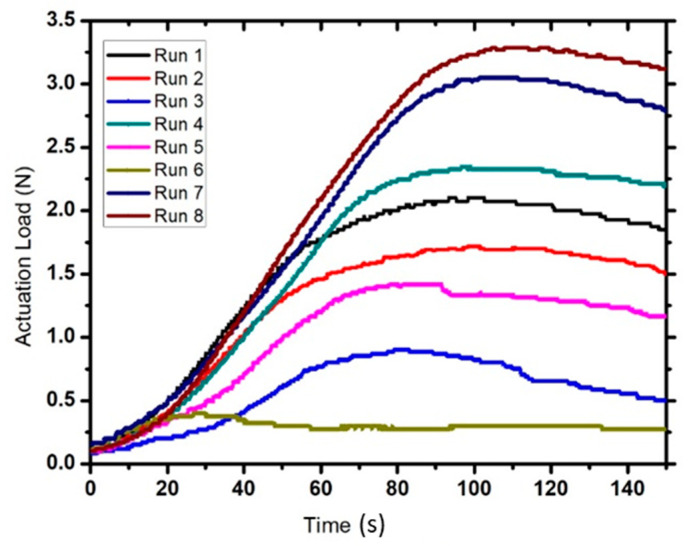
Actuation load curves for each experiment.

**Figure 10 materials-14-01264-f010:**
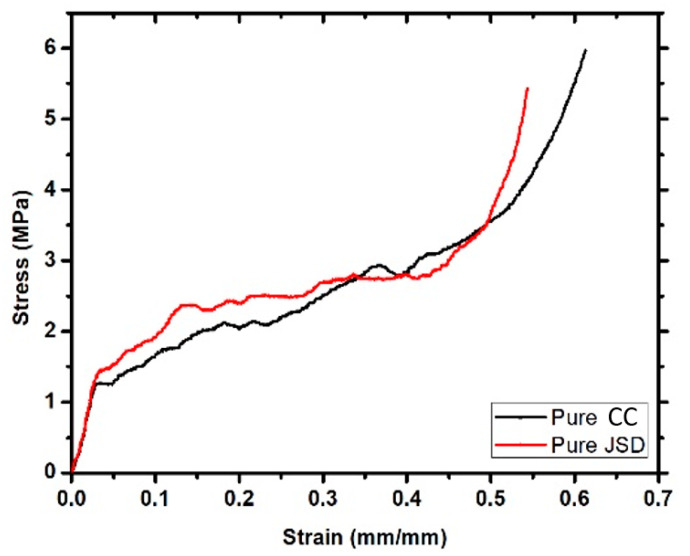
Stress strain graph for both Pure CC and pure JSD.

**Figure 11 materials-14-01264-f011:**
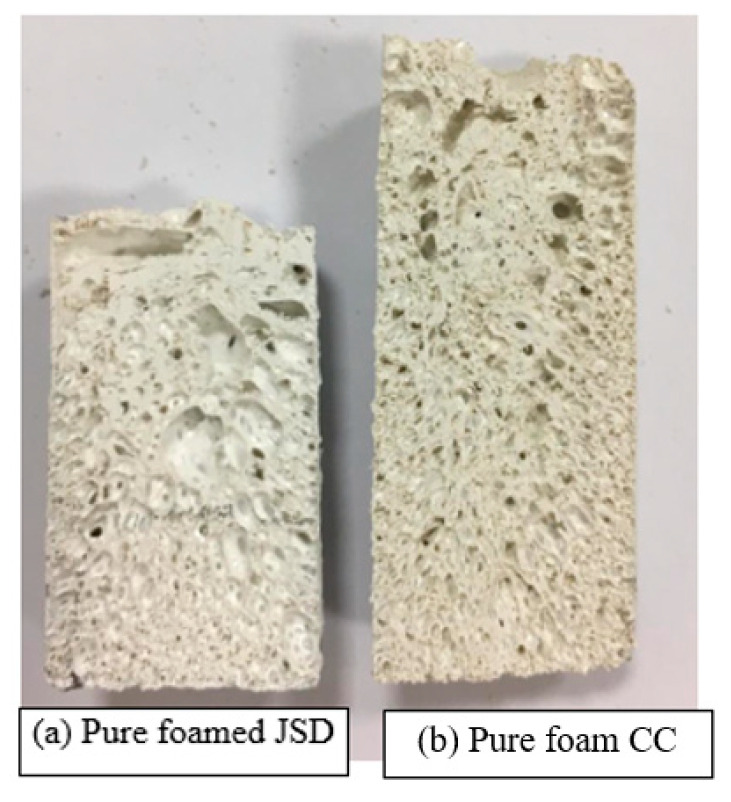
Foamed Samples.

**Table 1 materials-14-01264-t001:** The Taguchi Map design for this research. NPs: Nanoparticles; JAD: Jotun Super Durable 2903^®^; CC: Corro-Coat PE Series 7^®^.

	Parameters	NPs%	Polymer Type	Packing Pressure (lbs.)	Holding Time (Minutes)	Temperature (°C)	Foaming Time (Minutes)
Run #							
1		0	CC	7500	1	260	15
2		0	CC	7500	3	290	10
3		0	JSD	10,000	1	260	10
4		0	JSD	10,000	3	290	15
5		2	CC	10,000	1	290	15
6		2	CC	10,000	3	260	10
7		2	JSD	7500	1	290	10
8		2	JSD	7500	3	260	15

**Table 2 materials-14-01264-t002:** Average calculated values for recovery speed and time for each experiment.

Run#	1			2			3			4			5			6			7			8		
**Samples**	S1	S2	S3	S1	S2	S3	S1	S2	S3	S1	S2	S3	S1	S2	S3	S1	S2	S3	S1	S2	S3	S1	S2	S3
**Initial height**	25	25	25	24	24	24	15	15	15	18	18	18	28	28	28	26	26	26	12	12	12	14	14	14
**Average Recovery Time (seconds)**	243.66			236			186			188.66			319.33			337.66			170			166.33		
**Standard deviation**	6.79			11.86			3.74			3.29			9.177			3.68			10.98			8.178		
**Average Recovery Speed (mm/min)**	5.94			6.36			5.16			5.82			5.20			4.68			4.02			4.92		
**Standard deviation**	0.059			0.30			0.343			0.47			0.21			0.129			0.49			0.84		

**Table 3 materials-14-01264-t003:** Average measured actuation load values for each experiment.

Run#	1	2	3	4	5	6	7	8
Maximum Actuation Load (N)	2.36	1.66	0.91	2.35	1.46	0.45	3.01	3.35
Standard deviation	0.271	0.246	0.64	0.227	0.062	0.041	0.169	0.49

**Table 4 materials-14-01264-t004:** Minitab results for the t-test.

Two-Sample T for l40 vs. l20				
Sample Length	N	Mean	Standard Deviation	SE Mean
120	3	2.667	0.257	0.15
140	3	2.233	0.126	0.073
Difference = μ (l40) − μ (l20)				
Estimate for difference: 0.433				
90% CI for difference: (0.082, 0.785)				
t-Test of difference = 0 (vs ≠): t-Value = 2.63 *p*-Value = 0.058 DF = 4				
Both use Pooled StDev = 0.2021				

**Table 5 materials-14-01264-t005:** The effect of NP% on recovery speed (mm/minute).

	Polymer Type	CC	JSD	Average
NP%				
0%		5.97	5.5	5.735
2%		4.92	4.47	4.695
Average		5.445	4.985	5.215

**Table 6 materials-14-01264-t006:** The effect of packing pressure on recovery speed.

	Packing Pressure	7500	10,000	Average
Holding Time				
1		5.94	5.16	5.55
3		5.82	6.0	5.91
Average		5.88	5.58	5.73

**Table 7 materials-14-01264-t007:** Comparing this work to previous work.

Yield Strength (MPa)	Yield Strength (MPa) after Adding NPs	Polymer Name	Reference
3.9	6.7	PMMA	Zhang, et al. 2017 [27]
5.7	4.7	Polyurethane	Soto, et al. 2018
2.61	3.0	Polyimide	Weng, et al. 2018
56.26	75.93	Epoxy Polymer	Yu, et al. 2019 [28]
1.5	1.2	JSD	Our material
1.25	0.52	CC	

## Data Availability

The raw data required to reproduce these findings are available to download from: https://data.mendeley.com/datasets/trj68sjmmj/draft?a=3bb46d27-4495-4879-90b3-2c87d8f27813 (accessed 6 March 2021).

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
