# Peer review of "Optimization of Actuation Load and Shape Recovery Speed of Polyester-Based/Fe3O4 Composite Foams"

_materials, 2021, doi:10.3390/ma14051264_

Round 1
Reviewer 1 Report
In my opinion, this manuscript now can be accepted for publication.
Author Response
Thanks for recommending our work for puplication
Reviewer 2 Report
The paper is focused on the preparation and characterization of polyester-based polymers/Fe3O4 nanocomposite. The topic falls within the scope of the journal. I recommend its publication after the following revisions:
- Several typos are present in the text. Please check and revise.
- Introduction should be updated by highlighting that the incorporation of nanoparticles with variable shape can be effective to improve the mechanical properties of polymers allowing to fabricate nanocomposites functional for specific technological applications. Recent literature (doi.org/10.1016/j.clay.2019.105416; doi.org/10.1021/acs.macromol.6b02629) could be quoted to support this consideration.
- Table 2,3. Errors (determined as standard deviations) of the average values should be reported.
- References for the the T_test should be indicated.
- What are the sizes of the prepared Fe3O4 nanoparticles? Microscopic investigations on the nanoparticles as well as on the tablets could be interesting to investigate their morphological charcacteristics.
Author Response
Kindly refer to the attached file, it has a table that shows authors' response to the comments

Reviewer 3 Report
Few Minor points.
(1) Authors need to incorporate some interesting structural characterization data in the abstract part of the manuscript.
(2) Authors need to incorporate some crescent references related to introduction part of the manuscript to make it more interesting for the readers.
(3) All the data must be compared with previous reports in tabulated form to show superiority of the presented work.
(4) Authors need to include future prospective of the work in the conclusion part of the manuscript.
(5) English need to improve.
(6) Please include TGA and Powder Xray for composite foams.
(7) Please include FT-IR in the manuscript.
Author Response

(The authors gave the same response as above.)

Reviewer 4 Report
In this paper, polyester-based polymers/Fe3O4 nanocomposite foams were prepared in order to study their performance namely shape recovery speed and actuation load. The paper is well written; however, before publication the following improvements are recommended:
-the Introduction section it should be improved with a short presentation of the different types of foams.
-in many cases, a space must be left between the numerical value and the unit of measurement. In addition, the authors must be careful with the writing of the unit of measure (e.g. for density g/cm3->cm must be set to the 3rd power; see the beginning of section 4.2 and in the rest of the paper).
-the scale bar must be added to Figure 2.
-more information about the compression device must be added: the maximum value of the load-cell, testing conditions, loading speed, etc.
-the quality of figures 3, 6 and 7 needs to be improved.
-the standard deviations must be added to Tables 2 and 3.
-what is the reason for producing specimens with different heights (20 and 40 mm)?
-references must be added for the used equations.
-what is the purpose of the last average in tables 5 and 6 (the horizontal one)?
-for figure 7 the authors must add explanations regarding the three characteristic regions and properties resulting from the compression tests (typical of cellular materials). There are certain research groups (Fiedler et al., Linul et al., Movahedi et al., Orbulov et al., etc.) dealing with the physical and mechanical characterization of such cellular materials. Please refer to their works in the Discussion section.
-section 4 should present in-depth discussions and comparisons with the literature.
-the list of references is quite poor. Therefore, some recent developments published in the Materials Journal should be considered, showing a continuity between the present work and those reported in the literature on similar topics.
-English is not the native language of this Reviewer; however, the manuscript requires some corrections.
Author Response

(The authors gave the same response as above.)

Round 2
Reviewer 2 Report
The paper can be published in the present form.